

# HOM cluster decomposition in APi-TOF mass spectrometers

Tommaso Zanca[1], Jakub Kubečka[1], Evgeni Zapadinsky[1], Monica Passananti[1,2], Theo Kurtén[3], and Hanna Vehkamäki[1]

[1]Institute for Atmospheric and Earth System Research / Physics, Faculty of Science, University of Helsinki, P.O. Box 64, 00014 Helsinki, Finland
[2]Dipartimento di Chimica, Università di Torino, via Giuria 5, 10125 Torino, Italy
[3]Institute for Atmospheric and Earth System Research / Chemistry, Faculty of Science, University of Helsinki, P.O. Box 55, 00014, Helsinki, Finland

**Correspondence:** Tommaso Zanca (tommaso.zanca@helsinki.fi)

**Abstract.** Identification of atmospheric molecular clusters and measurement of their concentrations by APi-TOF mass spectrometers may be affected by systematic error due to possible decomposition of clusters inside the instrument. Here, we perform numerical simulations of decomposition in an APi-TOF and formation in the atmosphere of a set of clusters which involve a representative kind of highly-oxygenated organic molecule (HOM), with molecular formula $C_{10}H_{16}O_8$. This elemental

composition corresponds to one of the most common mass peaks observed in experiments on ozone-initiated autoxidation of $\alpha$-pinene. Our results show that decomposition is highly unlikely for the considered clusters, provided their bonding energy is large enough to allow formation in the atmosphere in the first place.

## 1 Introduction

Recent developments in mass spectrometry have brought huge advancements to the field of atmospheric science. For example,

mass spectrometers are now able to detect ppq-level concentrations of both clusters and precursor vapours in atmospheric samples (Junninen et al., 2010; Jokinen et al., 2012), as well as directly explore the chemistry of new-particle formation (NPF) in the atmosphere (Kulmala et al., 2014; Almeida et al., 2013; Bianchi et al., 2016; Ehn et al., 2014; Thomas et al., 2016; Hogan and de la Mora, 2010).

One of the most common mass spectrometers used to measure online cluster composition and concentration in the atmo-

sphere is the Atmospheric Pressure interface Time Of Flight Mass Spectrometer (APi-TOF MS). It can detect naturally charged molecules and clusters, or it can be used in combination with a Chemical Ionization chamber (CI-APi-TOF) to detect neutral molecules and clusters. Unfortunately, the detection process in the APi-TOF involves energetic interactions between the carrier gas and the clusters, possibly leading to their decomposition, and thus altering the measurement results. Specifically, the ions are guided and focused by an electric field inside the atmospheric pressure interface (APi) through a series of three vacuum

chambers before arriving to the time of flight mass spectrometer itself. The pressure decreases between successive chambers, until the final value of $10^{-6}$ mbar is reached in the TOF. Inside the APi, the clusters can collide with carrier gas molecules, and can possibly be decomposed in the process.



**Figure 1.** Molecular structure of the model $C_{10}H_{16}O_8$ HOM used in this study.

Cluster decomposition in the APi-TOF is one of the main sources of uncertainty in the measurements. Lack of accuracy for a quantitative estimate of decomposition for example makes it difficult to draw definitive conclusions on the presence (or absence) of certain molecular clusters in the atmosphere. Often the absence of observations of specific clusters by APi-TOF has led to speculation about decomposition inside the mass spectrometer (Olenius et al., 2013). Recently, a numerical model to study decomposition in the APi-TOF has been developed by Zapadinsky et al. (2018). The decomposition model has been tested on simple clusters involving one bisulfate anion and two sulfuric acid molecules, giving very good agreement with experimental results (Passananti et al., 2019).

The uncertainties on cluster concentration measurements by APi-TOF and the lack of a comprehensive understanding of decomposition inside the instrument are the motivations of the present work. Here we use a theoretical model to study in detail the decomposition of clusters involving so-called Highly-Oxygenated organic Molecules (HOM), which have recently been identified as a key contributor to NPF (Bianchi et al., 2019). HOM are molecules formed in the atmosphere from Volatile Organic Compounds (VOC). Some VOC with suitable functional groups can undergo an autoxidation process involving peroxy radicals, generating polyfunctional low-volatility vapors (i.e. HOM) that subsequently condense onto pre-existing particles. HOM thus contribute to Secondary Organic Aerosol (SOA), which constitutes a significant fraction of the submicron organic aerosol mass and is known to affect the Earth's radiation balance (Jimenez et al., 2009; Donahue et al., 2009; Hallquist et al., 2009). Recent chamber experiments have shown that NPF induced by multicomponent systems, such as sulfuric acid, ammonia and HOM, could correctly reproduce the NPF events observed in boreal forest (Lehtipalo et al., 2018).

Our study involves a specific kind of representative HOM ($C_{10}H_{16}O_8$) in the APi. This elemental composition corresponds to one of the most common mass peaks observed in experiments on ozone-initiated autoxidation of $\alpha$-pinene, which also fulfills the "HOM" definition of Bianchi et al. (2019). The precise molecular structure was adopted from Kurtén et al. (2016), and corresponds to the lowest-volatility structural isomer of the three $C_{10}H_{16}O_8$ compounds investigated in that study. The structure of the molecule is shown in Fig. 1.

The main scope of this work is to determine to what extent we are able to perform measurements of atmospheric cluster concentrations using APi-TOF mass spectrometers. More specifically, we want to determine whether decomposition can possibly be responsible for the lack of observations of some HOM-containing clusters in an APi-TOF. Basically, the formation





(a)

(b)

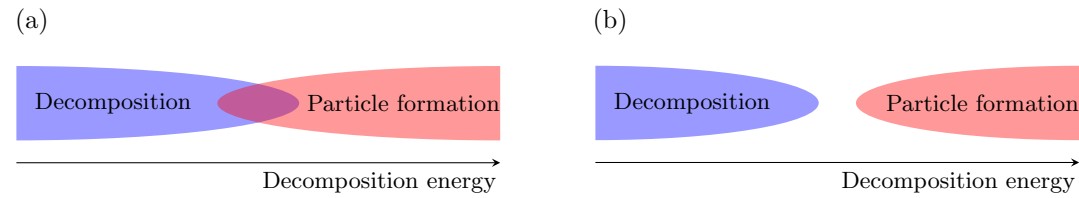

**Figure 2.** (a) The overlap between the decomposition and particle formation regions defines the range of decomposition energy at which both the events (decomposition in the APi-TOF and particle formation in the atmosphere) are allowed. (b) The two events are incompatible.

of clusters in the atmosphere (often referred to as "nucleation") is driven by the bigger stability of the cluster with respect to its separated molecular (and sometimes ionic) components (fragments). The degree of stability is given by the energy differ-

ence between the fragments and the cluster in their (electronic and rotational-vibrational) ground states, which is called either reaction, binding or decomposition energy, and it is equal to the amount of energy necessary to decompose the cluster (or, viceversa, the amount of energy released by the clustering). Higher decomposition energy implies lower decomposition and evaporation rates, and thus higher net formation rates in the atmosphere. On the contrary, decomposition in the APi-TOF is enhanced when the decomposition energy decreases, since less energy is required to break the cluster. It is clear at this point

that decomposition and new-particle formation ("nucleation") have opposite dependences on the cluster decomposition energy. For a given temperature and vapor concentration, two situations are possible (Fig. 2):

1. There is a limited range of values for the decomposition energy which allows both decomposition in the APi-TOF, and particle formation in the atmosphere.

2. The smallest decomposition energy that allows particle formation in the atmosphere is still too large to permit decompo-
sition in the APi-TOF.

Here, we predict both an upper bound for decomposition energy necessary for decomposition in the APi-TOF, and a lower bound for new-particle formation in the atmosphere given realistic vapor concentrations. The former can be evaluated using the numerical model developed by Zapadinsky et al. (2018), while the latter is computed using the Atmospheric Cluster Dynamics Code (ACDC) (McGrath et al., 2012). In the present work the simulations on decomposition have been performed by a new
C++ version of the decomposition code, which keeps the same basic algorithm but performs faster.

Another purpose of this work is to analyze the dynamics of complex clusters in the APi, showing where and why the decomposition events take place.

In Section 2 we present the decomposition model. In Section 3 we show the main results on HOM cluster decomposition and we compare them with ACDC simulations. Finally, in Section 4, we present the conclusions.





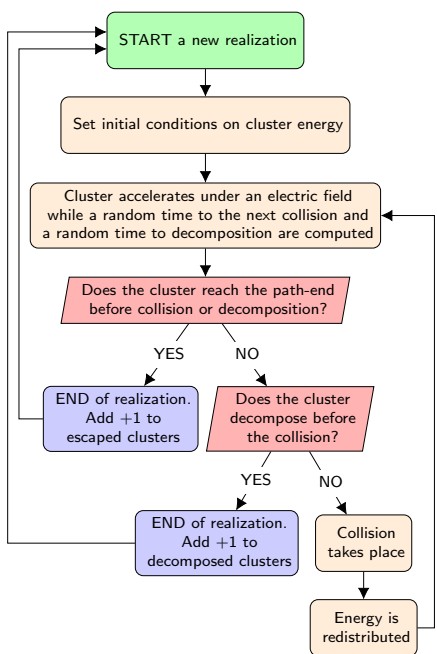

**Figure 3.** (Color online). Simplified flowchart of the decomposition code. The process is repeated up to the required number of realizations.

## 2 Method

The model computes the probability of decomposition of a single negatively-charged cluster from a large number ($\approx 10^3$) of independent stochastic realizations of its dynamics in the APi. The algorithm of the code is presented in the simplified flowchart shown in Fig. 3. We give only a brief description of the algorithm here, for full details see Zapadinsky et al. (2018). The dynamics starts with the ionized cluster accelerating in the mass spectrometer under the effect of an electric field generated by the electrodes inside the APi. While moving, a random time interval to the next collision is computed from the cumulative distribution function $\mathcal{P}_{\text{coll}}(t)$, which expresses the probability to encounter a collision after a time $t$:

$$\mathcal{P}_{\text{coll}}(t) = 1 - e^{-\int_0^t \Upsilon[v(t')]\,dt'} . \tag{1}$$

Here, $\Upsilon$ is the collision frequency which depends on the cluster velocity $v$, while $t = 0$ is the moment when the previous collision occurred. Simultaneously, another random time interval is computed for the decomposition event from a Poisson distribution, which corresponds to the time-dependent survival probability $\mathcal{P}_{\text{surv}}$ after collision:

$$\mathcal{P}_{\text{surv}}(t) = 1 - \mathcal{P}_{\text{dec}} = e^{-k(\Delta E)t} , \tag{2}$$

where $\mathcal{P}_{\text{dec}}$ is the decomposition probability and $k$, the decomposition rate constant, is the inverse of the statistical average of decomposition time, which depends on the cluster excess energy $\Delta E$ beyond its decomposition energy threshold. The decomposition time is interpreted as the time the cluster spends intact before decomposing. The decomposition rate constant is





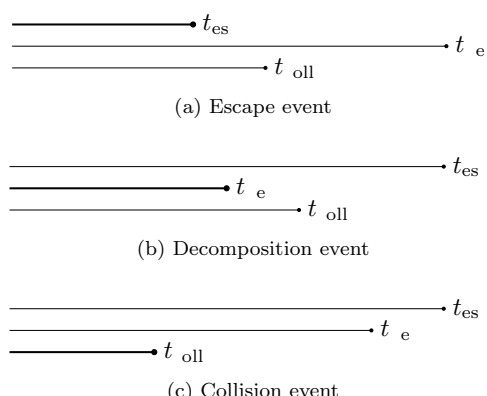

(a) Escape event

(b) Decomposition event

(c) Collision event

**Figure 4.** Three possible situations generated by the algorithm.

derived from Phase Space Theory of chemical reactions (PST). Basically, it is determined by the ratio of the densities of states of products to the densities of states of reactant, exploiting the detailed balance condition (Zapadinsky et al., 2018). This is a more rigorous derivation than RRKM theory, since the latter does not take into account angular momentum conservation in the decomposition process.

At this point, if the time required by the cluster to escape from the simulated region of the mass spectrometer is less than both the collision and decomposition times, the single realization is completed and the count of intact clusters is increased by one (Fig. 4a). A second possibility is that the decomposition time is smaller than collision and escape times, in which case the count of decomposed clusters is increased by one (Fig. 4b). Finally, as third possibility, the collision time may be the smallest, in which case a collision between the cluster and a carrier gas molecule takes place (Fig. 4c).

The dynamics of the collision is described by a stochastic process, where random velocities for the carrier gas molecules are computed from the Maxwell-Boltzmann distribution. In this case, the motion of the gas molecules is considered as solely translational. During the collision, the kinetic energy of the two colliding objects is partially transferred to the internal degrees of freedom of the cluster (vibrational and rotational modes), according to the microcanonical ensemble approach: any configuration of the system with the same energy is equally probable. The fundamental relation governing this statistics is given by the proportionality between the probability density function of energies for two different interacting degrees of freedom and their densities of states:

$$f_{a,b}(\epsilon_a, \epsilon_b) \propto \rho_a(\epsilon_a)\,\rho_b(\epsilon_b)\,\delta(E - \epsilon_a - \epsilon_b)\,, \tag{3}$$

where $\epsilon_i$ is the energy of mode $i$, $\rho_i$ is the density of states of mode $i$ and the Dirac delta function $\delta$ ensures conservation of total energy $E$. After integrating Eq. 3 over all the possible values that $\epsilon_b$ can assume, we get the probability density function for $\epsilon_a$:

$$f_a(\epsilon_a) = \frac{\rho_a(\epsilon_a)\,\rho_b(E - \epsilon_a)}{\int_0^E d\epsilon_a\,\rho_a(\epsilon_a)\,\rho_b(E - \epsilon_a)}\,. \tag{4}$$





Notice that this formula holds at equilibrium. This assumption is justified from the fact that the energy transfer takes place at time scales much shorter than the time scale between two consecutive collisions. Specifically, vibrational-vibrational energy exchange takes place in about $10^{-13}$–$10^{-12}$ s, rotational-vibrational exchange in $10^{-11}$ s and collisions every $10^{-9}$–$10^{-5}$ s.

After collision, the energy is then redistributed between rotational and vibrational modes following the same principle, and the dynamics continues with the acceleration of the cluster in the electric field.

In region IV, where the quadrupole is located, the cluster is subjected to additional acceleration in the transverse directions, because of the alternating electric field generated by the quadrupole. The dynamics is determined by the renowned Mathieu equation (Miller and Denton, 1986):

$$\frac{d^2y}{d\xi^2} + (a - 2q\cos 2\xi)y = 0 , \tag{5}$$

where

$$a = \frac{4eU}{mr_0^2\omega^2} , \quad q = \frac{2eV}{mr_0^2\omega^2} \quad \text{and} \quad \xi = \frac{\omega t}{2} , \tag{6}$$

with $y$ one of the two transverse displacements from the quadrupole axis, $t$ the time, $e$ the elementary charge, $U$ the DC component and $V$ the AC amplitude of the electric potential, $m$ the cluster mass, $r_0$ the half-distance between quadrupole rods, and $\omega$ the AC angular frequency.

The computation of the density of vibrational and rotational states requires the knowledge of the corresponding energy levels, which are computed by the quantum chemistry program Gaussian, using the PM7 semi-empirical method, which is the newest and generally best semi-empirical method available in Gaussian (Frisch et al., 2016). We note that while semi-empirical methods are unable to accurately model the energetics of molecular clustering, this is not a problem in the present study, as the binding/decomposition energy is treated here as a freely variable parameter (i.e. the unreliable PM7 binding energy is not actually used). The purpose of the PM7 optimizations and frequency calculations is simply to provide a qualitatively correct distribution of rotational and vibrational energy levels. As vibrational modes with wavenumbers above $2500\ \mathrm{cm}^{-1}$ are strongly underestimated by the PM7 method, we have further rescaled the frequencies following the relation suggested in Rozanska et al. (2014):

$$\begin{aligned}
\nu_{\mathrm{PM7}}^{\mathrm{scaled}} &= \nu_{\mathrm{PM7}}^{\mathrm{unscaled}} & \text{for } \nu < 125\ \mathrm{cm}^{-1}, \\
\nu_{\mathrm{PM7}}^{\mathrm{scaled}} &= 1.006\,\nu_{\mathrm{PM7}}^{\mathrm{unscaled}} - 27.7 & \text{for } 125\ \mathrm{cm}^{-1} < \nu < 2500\ \mathrm{cm}^{-1}, \\
\nu_{\mathrm{PM7}}^{\mathrm{scaled}} &= 3.99\,\nu_{\mathrm{PM7}}^{\mathrm{unscaled}} - 7849.2 & \text{for } \nu > 2500\ \mathrm{cm}^{-1}.
\end{aligned} \tag{7}$$

The vibrational density of states is then computed using the harmonic approximation, which is most likely the biggest source of error in the evaluation of the cluster survival probability. (See Section 3 for a sensitivity analysis on the effect of varying the vibrational frequencies).



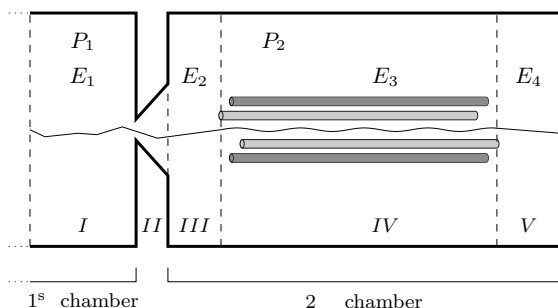

**Figure 5.** Simulated region of APi-TOF mass spectrometer. The dashed lines identify the regions with different voltages. The thin solid line illustrates a single realization of a cluster trajectory. The regions' lengths are not to scale.

## 3 Results

### 3.1 Decomposition in the APi

The simulation of decomposition involves only a portion of the total length of the APi-TOF mass spectrometer. Specifically, the simulations take place in the most critical region, ranging from the end of the first to the end of the second chamber of the APi, where the pressure values make decomposition possible: at the beginning of the first chamber the collisions are not energetic enough, while in the third chamber the carrier gas is so sparse that no collision happens (Zapadinsky et al., 2018).

In Fig. 5 we have sketched the simulated section of the instrument. We subdivide the simulated section into 5 regions with different longitudinal electric field and pressure values. Region I is at the end of the first chamber, region II defines the interface between the first and the second chambers and regions III, IV and V are located in the second chamber of the APi. The pressure in the first region is $P_1$, while in the regions III, IV and V it is $P_2$. In the region II, where the skimmer is located, the pressure changes continuously from $P_1$ to $P_2$. The electric field takes different values in the regions I, III, IV and V, while in II it is set to zero. The voltage and pressure configuration inside the APi chambers, used in the CLOUD10 experiments (Lehtipalo et al., 2018), is the following:

– $P_1 = 201$ Pa

– $P_2 = 2.96$ Pa

– $E_1 = 7189$ V/m

– $E_2 = 804.9$ V/m

– $E_3 = 18.04$ V/m

– $E_4 = 1104$ V/m

– $V_{DC} = 0$ V

– $V_{AC} = 200$ V



    – $\Omega = 1.3$ MHz

where $V_{\text{DC}}$ and $V_{\text{AC}}$ are the direct and alternating components of the quadrupole electric potential in region IV, while $\Omega$ is its radio frequency. We will use this voltage and pressure configuration in our simulations. It is important to notice here that the electric field takes very different values in different regions, varying over 3 orders of magnitude, which will greatly diversify
the probability of decomposition at different locations.

    In this study the cluster decomposition energy $E_{\text{f}}$ is treated as a free parameter. This allows us to explore the behaviour of cluster survival probability as a function of $E_{\text{f}}$ over a large energy range. Moreover, keeping $E_{\text{f}}$ as a variable is also useful since its computation by quantum chemistry calculations (especially low-level methods such as PM7 used here) is affected by large errors.

The clusters studied here are formed by one bisulfate anion, one to two sulfuric acid molecules and one to three HOM molecules. The clusters are assumed to decompose by losing one HOM, as follows:

$$(\text{HSO}_4^-)(\text{H}_2\text{SO}_4)_n(\text{HOM}_{10})_m \rightarrow (\text{HSO}_4^-)(\text{H}_2\text{SO}_4)_n(\text{HOM}_{10})_{m-1} + (\text{HOM}_{10}) \tag{8}$$

with $n = 1, 2$, $m = 1, 2, 3$ and $\text{HOM}_{10}$ is the structure shown in Fig. 1. Our specific choice for HOM is not unique: this is one among many potential HOM produced in the $\alpha$-pinene + $O_3$ reaction. The compound chosen is broadly representative of
autoxidation products as it contains both hydroperoxide, ketone and carboxylic acid groups. The clusters have been constructed by first maximizing H-bonds between the $\text{HSO}_4^-$ core ion and other molecules, and then maximizing other H-bonds without creating too much strain. We note that the cluster conformers obtained in this fashion are unlikely to correspond precisely to the global energy (or free energy) minima. However, this mainly affects the computed binding energy – which (as pointed out earlier) is not actually used in this study. For purposes of generating a representative ensemble of vibrational and rotational
energy levels, the conformer generation approach used here is adequate.

    For each kind of cluster we performed several simulations at different $E_{\text{f}}$. The simulated carrier gas is air, modelled to consist of 80% nitrogen and 20% oxygen.

    The final survival probabilities $P_{\text{surv}}$ are shown in Fig. 6. As expected, $P_{\text{surv}}$ is monotonic with respect to $E_{\text{f}}$. From these results we can identify a boundary at $E_{\text{f}} \approx 25$ kcal/mol beyond which decomposition is highly unlikely. The weakest bonds
between non-hydrogen atoms in HOM molecules are likely to be the $O-O$ bonds of peroxide or hydroperoxide groups. These typically have bond dissociation energies around $35 - 50$ kcal/mol (Bach et al., 1996; Schweitzer-Chaput et al., 2015, 2017). Vinyl peroxide systems have much lower dissociation energies, but (with the exception of short-lived vinyl hydroperoxides generated in ozonolysis) these are unlikely to form in gas-phase oxidation processes. The threshold cluster decomposition energies reported here are thus much lower than the energies required to dissociate even the weakest covalent bonds in the
HOM molecules.

    Larger clusters are found to decompose more easily. This can be understood looking at the probability density function (PDF) of vibrational energy at equilibrium at temperature $T = 300$ K (Fig. 7a). The distributions have been computed multiplying the



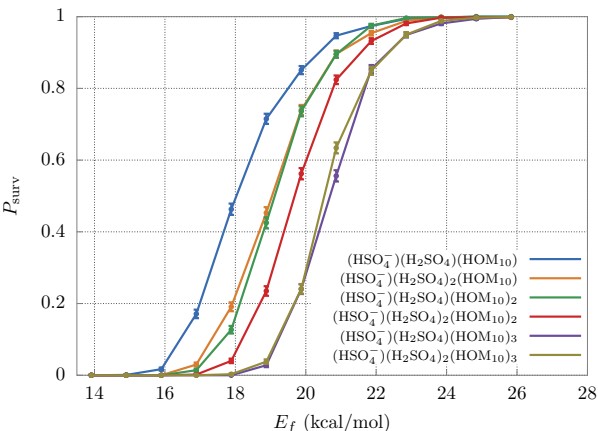

**Figure 6.** (Color online). Survival probabilities $P_{\mathrm{surv}}$ for different HOM clusters as a function of their decomposition energy $E_{\mathrm{f}}$. The error bars are relative only to statistical error.

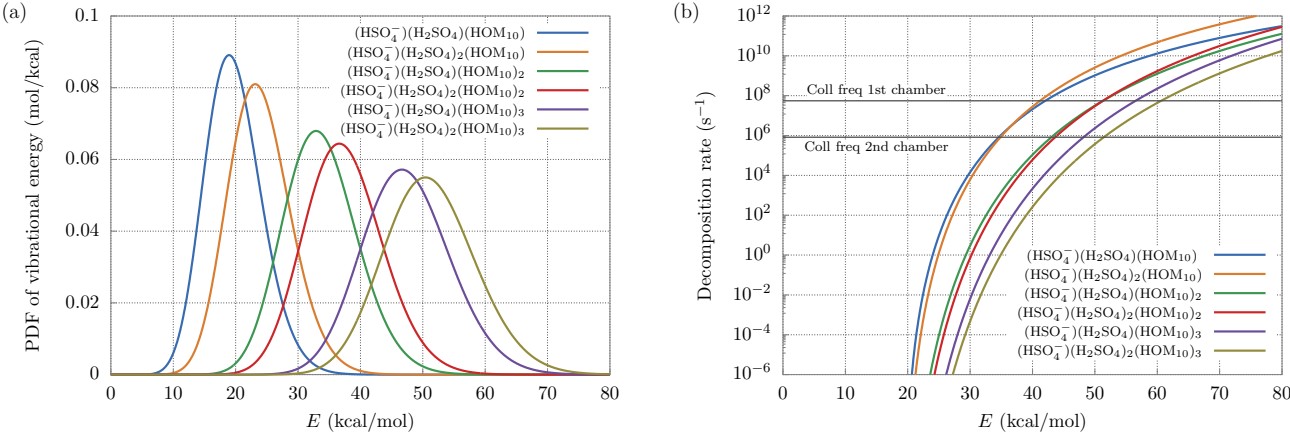

**Figure 7.** (Color online). (a) Distribution of vibrational energy at equilibrium at temperature $T = 300$ K. (b) Decomposition rate constants at $E_{\mathrm{f}} = 20\ \mathrm{kcal/mol}$. Collision frequencies in the first and second chambers are shown as reference.

density of states of vibrational modes by the Boltzmann factor, and then renormalized:

$$\mathrm{PDF}(E) = \frac{\rho_v(E)\, e^{-E/k_B T}}{\int_0^{+\infty} \rho_v(E)\, e^{-E/k_B T}} \ .\tag{9}$$

Larger clusters contain a higher number of bonds, which for fixed decomposition energy, will each increase the internal energy of the cluster, because of the equipartition theorem. This leads to a higher probability to exceed the decomposition energy, even at equilibrium condition, i.e. in the absence of an electric field. Suppose, for example, $E_{\mathrm{f}} = 20\ \mathrm{kcal/mol}$. Fig. 7a shows that the smallest cluster has roughly 50% probability to have an internal energy higher than $E_{\mathrm{f}}$, while the biggest cluster is certainly at higher energy than $E_{\mathrm{f}}$, with an average near $50\ \mathrm{kcal/mol}$. This effect is in part counterbalanced by the behaviour



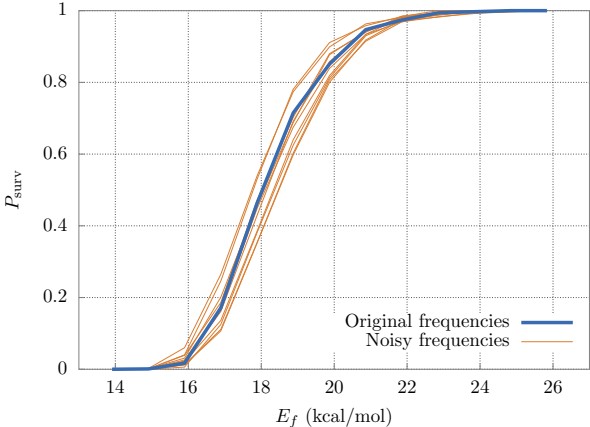

**Figure 8.** (Color online). Deviations (thin lines) from original survival probability $P_{surv}$ (thick line) for 10 different sets of "noisy" input frequencies. The new sets of frequencies have been constructed introducing a 20% error on original values.

of decomposition rate constant as a function of cluster size. As we can see in Fig. 7b, the decomposition rate at fixed internal energy tends to be smaller for larger clusters. Indeed, a larger number of bonds reduces the probability to find a large quantity of energy in a single bond.

The model requires as input parameters the vibrational and rotational frequencies of both the reactant and products from an external quantum chemistry program (in our case Gaussian 16). These input data are affected by errors that depend on the level of accuracy of the computational method. It is therefore interesting to study how errors in the inputs affect the survival probability. For this reason, we generated new random frequencies from normal distributions centered at the original frequencies, with a standard deviation equal to 20% of their values. Fig. 8 shows the deviations of the survival probability for 10 different sets of input frequencies. Here we can see that, even with large errors in the input parameters (20%), the final results change by no more than 10%. These results demonstrate the low sensitivity of the model on deviations in the input frequencies, thus validating our use of rather crude approaches for both the quantum chemical calculations, and the conformer generation.

Let us now analyze the locations where collisions take place in the APi. In particular, we are interested in the highest-energy collisions, which lead to decomposition. We dub them as *fatal* collisions. For this study we used the voltage configuration and pressures used in CLOUD10 and we collected data from 5000 realizations of $(HSO_4^-)(H_2SO_4)(HOM_{10})$ cluster dynamics at $E_f = 17.88$ kcal/mol, corresponding to a survival probability $P_{surv} \approx 50\%$ (see Fig. 6).

Fig. 9a shows the average number of total and fatal collisions per single cluster evolution in different regions of the APi. The normalized number of fatal collisions (dark bars in Fig. 9a) can be interpreted as the probability of the cluster to decompose in the correspondent region in a complete realization. Summing up these values of all five regions gives the total probability of decomposition. Fig 9b instead shows the ratio between fatal and total collisions, which can be interpreted as the probability





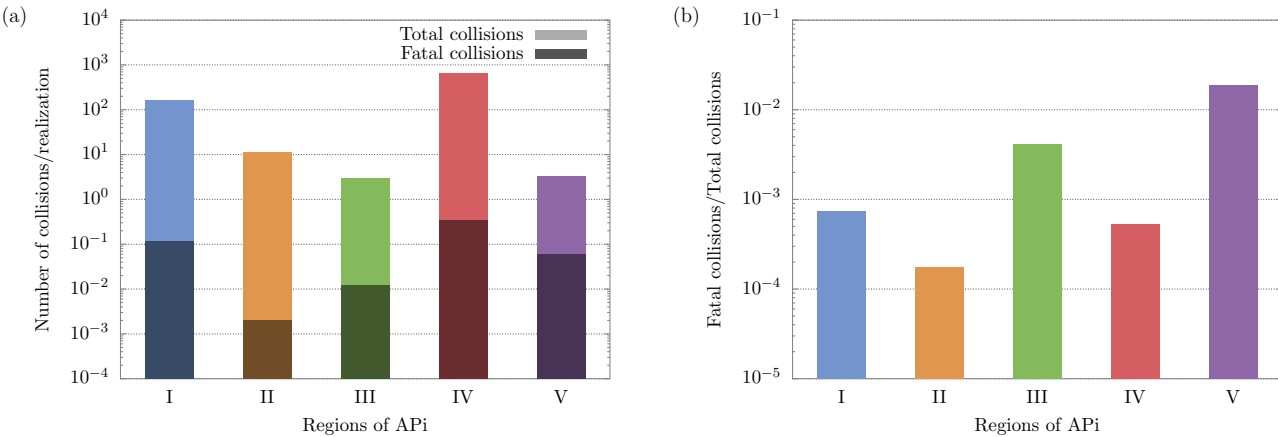

**Figure 9.** (Color online). (a) Average number of total and fatal collisions per single cluster evolution in the different APi regions. (b) Ratio between total and fatal collisions. The results have been obtained from 5000 realizations of $(HSO_4^-)(H_2SO_4)(HOM_{10})$ cluster dynamics. Notice that regions' lengths of the APi are different, in particular region IV, where the quadrupole is located, is much longer than the other regions, which leads to a higher number of collisions.

of the cluster to decompose after experiencing a collision in the correspondent region. Therefore, Fig. 9b identifies the regions where the collisions are most energetic (regions III and V).

The cluster dynamics starts in the first region (end of first APi chamber), where the high pressure (201 Pa) generates a large number of collisions ($\approx 200$ collisions/realization). Here the ratio between fatal collisions and total collisions is about 0.07% (Fig. 9b), a sign of low-energy collisions despite the very strong electric field (7189 V/m). When the cluster moves to the region II, in the skimmer, we see a drop in collisions, reflecting the negative pressure gradient. Moreover, the absence of electric field in this region makes fatal collisions very unlikely to happen ($\approx 0.017\%$ probability). In the region III, at the beginning of the second APi chamber, the pressure is low (2.96 Pa), and the cluster experiences few collisions. However, the strong electric field (804.9 V/m) leads to cluster acceleration and an increase of fatal collisions, about 0.4% of total collisions. The passage to region IV is characterized by a high speed of the cluster acquired in the previous region. This leads to a large number of decompositions, as shown in Fig. 9a. Subsequently, the cluster decelerates because of the very weak electric field in this region (18.04 V/m). Since the cluster moves for a long distance at low velocities in region IV, the average probability of decomposition per collision drops to $\approx 0.05\%$. Finally, in the region V, the cluster accelerates again due to a strong electric field (1104 V/m), which is visible from the large increase of the probability of fatal collisions ($\approx 2\%$). The cumulative survival probability of clusters as they travel through regions I to V is illustrated by the red dots in Fig. 10. The decomposition probabilities in each respective region are indicated by the histogram bars.

Although the decomposition is mainly induced by the voltages applied between regions II and III as shown in Passananti et al. (2019), these results demonstrate that the decomposition takes place mainly in region IV (Fig. 9a). This is due to the large acceleration acquired by the cluster in region III, with subsequent possible collision at high energy in the beginning of the next



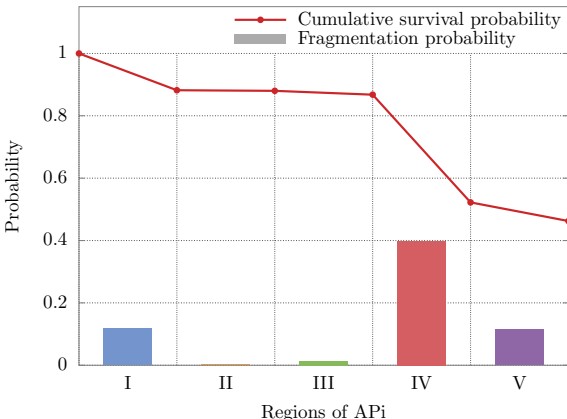

**Figure 10.** (Color online). Cumulative survival probability after going through each region (red dots) and decomposition probability (bars) for a single cluster realization. The decomposition probability in each region is given by the ratio of clusters decomposing inside the region, over clusters entering the region.

region. These results are complementary to previous data reported in literature (Passananti et al., 2019; Lopez-Hilfiker et al., 2016) and help to define the terms "declustering voltages" as the voltages that induce decomposition (between the region II and III) and the "declustering region" as the region of the APi where the decomposition takes place (region IV).

In region IV a quadrupole is used as ion guide in order to focus the cluster trajectories, through an alternating transverse electric field (Miller and Denton, 1986). For this reason, the cluster is also subjected to a transverse acceleration in addition to the longitudinal one, increasing the velocity and, consequently, the probability of decomposition. In order to separate the contributions to decomposition from the longitudinal and the transverse electric fields, we performed additional simulations setting the quadrupole electric potential $V_{AC} = 0$. In this situation the motion in the transverse directions is negligible (as in the other regions of the APi). The increase in the final survival probabilities can be seen in Fig. 11a, which shows a minor deviation from the case with $V_{AC} = 200$ V. More specifically, Fig. 11b shows the slight reduction of decomposition probability within the quadrupole region when $V_{AC} = 0$. Hence, with the present voltage configuration, the quadrupole voltage does not affect the probability of decomposition significantly.

### 3.2 Formation in the atmosphere

In parallel to the simulations on decomposition in the APi, we used the ACDC code (McGrath et al., 2012) to compute the concentrations of $HOM_{10}$ needed to provide a significant enhancement of new-particle formation rate in the atmosphere (details on ACDC simulations provided in Appendix A). Specifically, at given $H_2SO_4$ monomer concentration and decomposition energy $E_f$, we computed the concentration of $HOM_{10}$ at which the new-particle formation rate is increased by $1\ \mathrm{cm}^{-3}\mathrm{s}^{-1}$ with respect to the pure sulfuric acid system (i.e. $J_{sa+hom} = J_{sa} + 1\ \mathrm{cm}^{-3}\mathrm{s}^{-1}$).





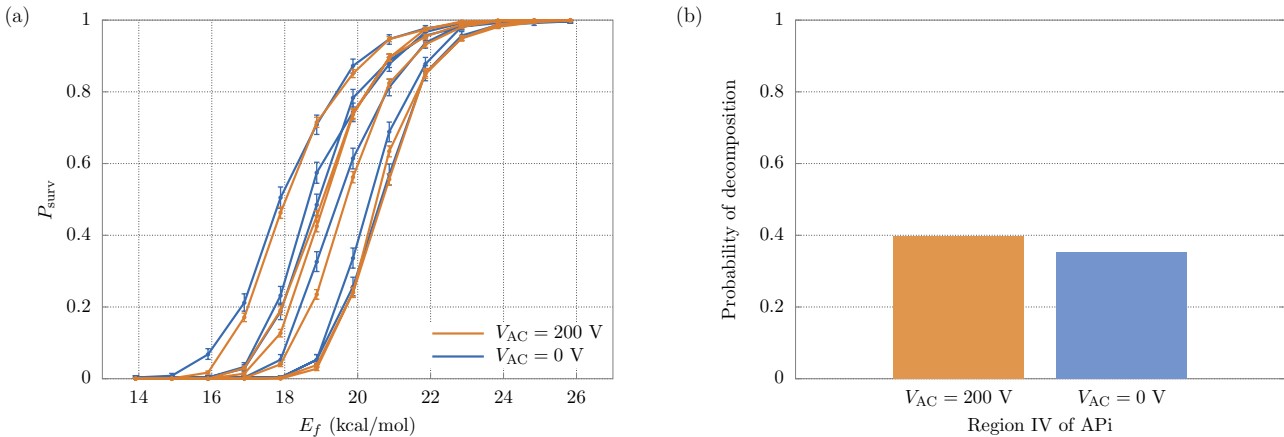

**Figure 11.** (Color online). (a) Comparison between survival probabilities in presence ($V_{AC} = 200$ V) and absence ($V_{AC} = 0$ V) of the quadrupole electric field. (b) Probability of decomposition in the quadrupole region, with and without quadrupole electric field.

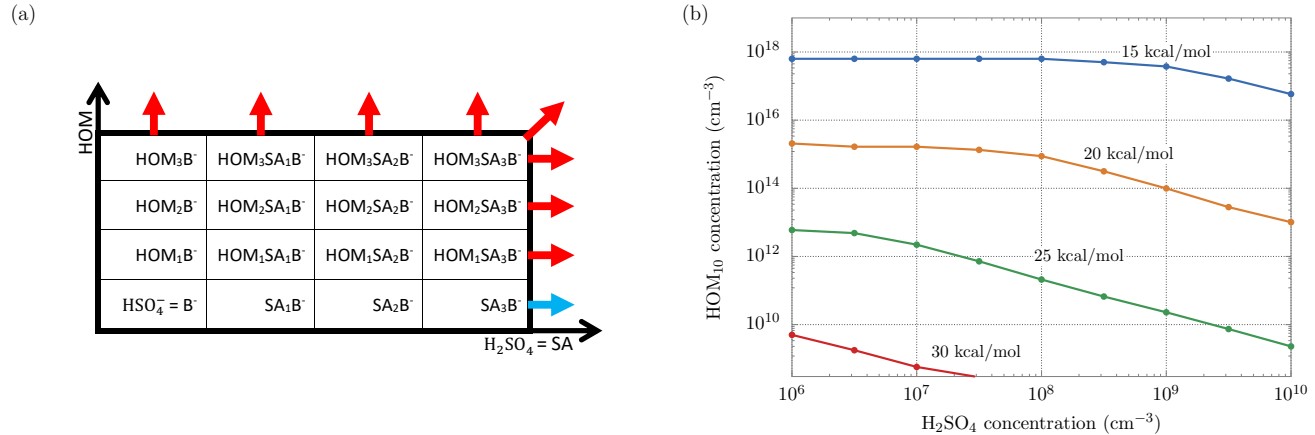

**Figure 12.** (Color online). (a) Clusters involved in the ACDC simulations. Out of the box, the clusters are considered to be stable and contributing to new-particle formation. The blue arrow indicates the flux of the outgrowing cluster that contributes to the formation rate of only sulfuric acid particles ($J_{sa}$). Adding $HOM_{10}$ monomers to the system, new outgrowing HOM clusters (red arrows) are participating in new-particle formation, resulting in a total formation rate $J_{sa+hom}$. (b) Concentrations of $HOM_{10}$ needed to increment the new-particle formation rate by $1 \ cm^{-3}s^{-1}$, for different decomposition energies.

In the simulation, the formation rate $J_{sa+hom}$ corresponds to clusters growing out of $(HSO_4^-)(H_2SO_4)_4(HOM_{10})_{0-3}$ or $(HSO_4^-)(H_2SO_4)_{0-3}(HOM_{10})_4$, as shown in Fig. 12a. The increment in the formation rate value here is indicative, and it serves as a reference for a reasonable NPF process in the atmosphere (Lehtipalo et al., 2018). The concentration of $HSO_4^-$ is kept constant at $700 \ cm^{-3}$, which corresponds to the steady-state concentration at the ions formation rate $J_{ions} = 4 \ cm^{-3}s^{-1}$, reproducing the galactic cosmic rays (GCR) conditions (Kirkby et al., 2016).



The HOM clusters are subsequently formed by collisions between bisulfate anions, sulfuric acid and HOM molecules. As we can see in Fig. 12b, the $H_2SO_4$ and $HOM_{10}$ concentrations needed to provide the increment $\Delta J = 1$ cm$^{-3}$s$^{-1}$ decrease when the decomposition energy increases, reaching experimental conditions (e.g. non-nitrate $HOM_{10}$ and $H_2SO_4$ concentrations measured in CLOUD experiment (Lehtipalo et al., 2018)) at $E_f > 30$ kcal/mol, where $HOM_{10}$ and $H_2SO_4$ concentrations do not exceed $10^8$ cm$^{-3}$. At this decomposition energy, decomposition in APi is negligible ($P_{surv} \approx 1$ in Fig. 6). Thus we can conclude that for the case of SA-HOM clusters, rapid formation in the atmosphere (given typical vapor concentrations) and significant decomposition in the APi are mutually incompatible situations (case b in Fig. 2).

## 4    Conclusions

In this work, we have presented the numerical results on decomposition inside an APi-TOF instrument of a specific class of atmospheric clusters that involve sulfuric acid and HOM molecules. A previously reported low-volatility $\alpha$-pinene ozonolysis product, with the molecular formula $C_{10}H_{16}O_8$, was used as a representative HOM.

There are three main results from our simulations. First, decomposition of HOM clusters in the APi requires a range of cluster decomposition energies which is incompatible with efficient cluster formation in the atmosphere given sub-ppb vapor concentrations. This result has been obtained by computing the cluster survival probability in the APi as a function of its decomposition energy, and then comparing the range of energies leading to decomposition with those needed to obtain a reasonably high new-particle formation rate in atmospheric conditions. The two ranges have no overlap – the highest energy allowing decomposition differs from the lowest energy allowing atmospheric new-particle formation by roughly 10 kcal/mol. Observations of SA-HOM clusters in CLOUD experiments (Lehtipalo et al., 2018) validate the results of our model.

Our second main result is the identification of the locations of the highest-energy collisions that lead to decomposition of SA-HOM clusters in the APi. The simulation shows that they are mainly localized in the quadrupole region (IV). This is due to the large velocity of the cluster acquired in the previous region (III), caused by the strong electric field. Moreover, our results show that the alternating field of the quadrupole in region IV has only a minor effect on decomposition probability.

As a third main result, we have shown that the model displays low sensitivity to changes in cluster vibrational and rotational frequencies. This feature allows us to obtain reliable results also with low-level input data (e.g. semi-empirical methods and limited configurational sampling).

This study was performed for a small set of HOM clusters with particular functional groups (e.g. hydroperoxides, ketones), but in future similar simulations could be performed for other types of clusters, for example to assess whether they should be detectable in an APi-TOF or not, or whether they could possibly affect new-particle formation, despite not being directly detected.



**Appendix A: ACDC simulation details**

The configurational sampling of the $H_2SO_4$ molecule and molecular clusters $(HSO_4^-)(H_2SO_4)_{0-3}$ has been performed as described by Kubečka et al. (2019) to obtain their thermodynamic ($T = 300$ K) properties at a high level of theory (DLPNO-CCSD(T)/aug-cc-pVTZ//$\omega$B97X-D/6-31++G(d,p) (Kubečka et al., 2019; Myllys et al., 2016; Riplinger and Neese, 2013; Riplinger et al., 2013, 2016; Chai and Head-Gordon, 2008; Dunning, 1989; Kendall et al., 1992; Liakos et al., 2015)).

The formation Gibbs free energies were used as input for the Atmospheric Cluster Dynamic Code (McGrath et al., 2012), which uses them to obtain cluster stabilities (i.e., evaporation rates). Bisulfate ions $HSO_4^-$ are in the atmosphere formed by collisions of "generic" anions and sulfuric acid $H_2SO_4$ (i.e., as $H_2SO_4$ is practically the strongest acid in the atmosphere, collision of $H_2SO_4$ with almost any anion will tend to produce $HSO_4^-$).

In the simulation, we assume a constant concentration of bisulfate ions equal to $700\,\mathrm{cm^{-3}}$, which approximately corresponds to the steady-state anion concentration at a representative total ion pair formation rate due to galactic cosmic rays (GCR) of $J_{\mathrm{ions}} = 4\,\mathrm{cm^{-3}s^{-1}}$ (Kirkby et al., 2016). We assume that formation of $(HSO_4^-)(H_2SO_4)_4$ (outgrowing cluster) or larger clusters leads to irreversible new-particle formation. Thus, at a specific $H_2SO_4$ monomer concentration, we simulated new-particle formation of sulfuric acid aerosols until steady-state had been reached. For each monomer concentration, the new-particle formation rate $J_{\mathrm{sa}}$ were noted down. (For simplicity, we neglect losses due to ion-ion recombination, wall-loss, dilution or coagulation, as our focus is on comparing the relative effect of HOM.)

As described in the manuscript of Kubečka et al. (2019), we performed 'weak' configurational sampling of molecules $HOM_{10}$, $H_2SO_4$ and molecular clusters $(HSO_4^-)(H_2SO_4)_{0-3}(HOM_{10})_{0-3}$ to obtain representative thermodynamic properties of these clusters at low level of theory (PM7 (Stewart, 2013)). By 'weak' configurational sampling we mean that just 1 conformation of $HOM_{10}$ is included for generating of HOM-containing clusters. However, as only the rotational and vibrational energy levels from the PM7 calculations are actually used, and not the electronic energies, this approach is sufficient for our purposes. We further assume that the evaporation rate of $H_2SO_4$ from $(HSO_4^-)(H_2SO_4)_x(HOM_{10})_{1-3}$ clusters is identical to the evaporation rate of $H_2SO_4$ from the previously calculated corresponding $(HSO_4^-)(H_2SO_4)_x$ clusters. The evaporation rate $\gamma$ of $HOM_{10}$ from $(HSO_4^-)(H_2SO_4)_x(HOM_{10})_y$ is calculated as

$$\gamma = \beta \cdot \frac{p_{\mathrm{ref}}}{k_B T} \cdot e^{\Delta G/k_B T} , \tag{A1}$$

where $k_B$ is Boltzmann's constant, $T$ is temperature, $\beta$ is the collision rate of $HOM_{10}$ and $(HSO_4^-)(H_2SO_4)_x(HOM_{10})_{y-1}$, $p_{\mathrm{ref}}$ is the reference pressure, and $\Delta G$ is the free energy change of the reaction

$$HOM_{10} + (HSO_4^-)(H_2SO_4)_x(HOM_{10})_{y-1} \longrightarrow (HSO_4^-)(H_2SO_4)_x(HOM_{10})_y .$$

$\Delta G$ is calculated as

$$\Delta G = \Delta G^{\mathrm{PM7}} - \Delta E_{\mathrm{el}}^{\mathrm{PM7}} + E_{\mathrm{f}} = \Delta G_{\mathrm{therm\&ZPE}}^{\mathrm{PM7}} + E_{\mathrm{f}} , \tag{A2}$$

where $E_{\mathrm{f}}$ is the parametrized value of the decomposition energy (varied between 15 and 30 kcal/mol as shown), $\Delta G^{\mathrm{PM7}}$ and $\Delta E_{\mathrm{el}}^{\mathrm{PM7}}$ are the reaction Gibbs free energy and electronic energy (respectively) computed at the PM7 level of theory, and



$\Delta G_{\text{therm\&ZPE}}^{\text{PM7}}$ corresponds to the thermal (and zero-point vibrational) correction to the reaction Gibbs free energy at the PM7 level.

Again, in the same way as in the decomposition model, we used the evaporation rates as input for the ACDC program assuming $(\text{HSO}_4^-)(\text{H}_2\text{SO}_4)_4(\text{HOM}_{10})_{0-3}$ or $(\text{HSO}_4^-)(\text{H}_2\text{SO}_4)_{0-3}(\text{HOM}_{10})_4$ to be the outgrowing clusters. Next, at given $\text{H}_2\text{SO}_4$ monomer concentration, and for a given value of $E_{\text{f}}$, we searched for the $\text{HOM}_{10}$ concentration at which the new-particle formation rate is increased by $1\,\text{cm}^{-3}\text{s}^{-1}$ (i.e., $J_{\text{sa+hom}} = J_{\text{sa}} + 1\,\text{cm}^{-3}\text{s}^{-1}$). The resulting graph is shown in the main article in Fig. 12b.

The above computations involved the following programs: the ABCluster program (Zhang and Dolg, 2015, 2016), the GFN2-$x$TB program (Grimme et al., 2017; Bannwarth et al., 2019), the Gaussian 16 program Revision A.03 (Frisch et al., 2016), the GoodVibes program (Funes-Ardois and Paton, 2016) and the Orca program version 4.0.1.2 (Neese, 2012).

*Author contributions.* TK and HV conceived the project. TZ and EZ wrote the code of the decomposition model. TZ performed simulations of the decomposition model, carried out the data analysis and wrote the first draft of the manuscript. JK performed simulations of the ACDC
code, produced the relative figures and wrote the supplementary information on ACDC simulations. MP helped in planning the project and interpreting the results in relation to experimental data. All authors contributed to writing the manuscript.

*Competing interests.* The authors declare that they have no conflict of interests.

*Acknowledgements.* We acknowledge the ERC Project 692891-DAMOCLES, the ATMATH Project and the Academy of Finland for funding, and the CSC-IT Center for Science in Espoo, Finland, for computational resources. We would like to thank Federico Bianchi and Siegfried
Schobesberger for valuable discussions, and Chao Yan for providing CLOUD campaign data.



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
