# Peer review of "HOM cluster decomposition in APi-TOF mass spectrometers"

_Atmospheric Measurement Techniques, 2019_

## Referee Comment (RC1) · Anonymous Referee #1 · 13 Mar 2020

Zanca et al. present numerical simulations of the decomposition of molecular clusters in Atmospheric Pressure interface Time of Flight (APi-ToF) mass spectrometers. APi-ToF has recently evolved into one of the most commonly used techniques to measure the concentration and molecular composition of both neutral and naturally charged atmospheric clusters. In the present manuscript, the potential decomposition of clusters that include highly-oxygenated organic molecules (HOMs) representative of a mass peak commonly observed in ozonolysis of alpha-pinene is studied using a numerical decomposition model that has been developed and previously tested on simple clusters by the same group. Thus, the manuscript contributes essential information for prudent interpretation of mass spectra of molecular clusters in a large number of new particle formation studies, and it is clearly within the scope of AMT. I cannot comment on all the

technical details of the manuscript but the presented work appears to be well-founded and rigorous. I recommend the following minor revisions for the authors to consider:

p. 5, line 87: The term "RRKM theory" is not explained. Please add a brief explanation.

p. 5, Figure 4: In my opinion, Figure 4 is not necessary. Please consider removing Figure 4.

p. 7/8: Lines 148-156 might be more clearly arranged in a table.

p. 10, Figure 8: Please explain in slightly more detail what you mean by '10 different sets of "noisy" input frequencies'.

p. 10/11, Figure 9a: I don't fully understand the dark bars presented in Figure 9a. While the figure caption suggests that these indicate the average number of fatal collisions, the main text (line 212) suggests that these are the "normalized number of fatal collisions". Please clarify.

p. 12, Figure 10: In the figure legend, change "Fragmentation probability" to "Decomposition probability".

p. 13, Figure 11a: What is represented by the different curves shown in Figure 11a? The colors 'blue' and 'orange' indicate the presence or absence of the quadrupole electric field but what is represented by the set of curves?

---

## Referee Comment (RC2) · Anonymous Referee #2 · 21 Apr 2020

In the present manuscript, numerical simulations of the decomposition of clusters in an APi-TOF mass spectrometer are performed, which are a representative type of a highly oxygenated organic molecule (HOM) with the molecular formula C10H16O8. This elemental composition corresponds to one of the most frequent mass peaks observed in experiments on ozone-initiated autoxidation of biogenic hydrocarbons. The results of the authors show that a decomposition of the considered clusters in the measuring instrument is highly unlikely, as long as their binding energy is large enough to enable their formation in the atmosphere in the first place. In general, HOMs have been at the center of atmospheric nucleation research for several years, especially since their molecular size and thus their vapor pressure make them one of the few organic compounds that can play a role in the early stages of particle formation. Their measurement is mainly performed with the Api-ToF MS also used in the manuscript, the reliability of the results produced by these measurements is therefore highly relevant and a publication in AMT is appropriate. The model used is highly specialized and cannot be easily reproduced in detail by experimentally educated scientists, but the results appear solid and well-founded. The model is described in sufficient depth and applied to the conditions of the mass spectrometer used. An error analysis is also part of the presented results. Finally, the results are related to atmospheric nucleation by simulating the conditions under which nucleation rates and concentrations correspond to those in the atmosphere and the CLOUD chamber. The manuscript is well written and generally clearly presented. Previous work is adequately cited. Therefore, I suggest to publish the manuscript in Atmospheric Measurement Techniques after considering the following minor comments and suggestions.

Figure 12 is a central figure in the manuscript: It would be desirable if this figure was discussed more intensively and in more detail in the text. Other illustrations, such as Fig. 8, are less meaningful to be shown graphically and could be deleted without loss of quality.

---

## Author Comment (AC1) · 3 May 2020

Dear Referee #1,

thank you for your comments. Here are our answers:

Q: p. 5, line 87: The term "RRKM theory" is not explained. Please add a brief explanation.

A: The RRKM theory was mentioned in the text because it presents some similarities with the actual approach of this manuscript for the derivation of the fragmentation rate. Nevertheless, in the present work we are using a different method, so we decided to avoid mentioning RRKM theory in order to avoid confusion among the readers.

Q: p. 5, Figure 4: In my opinion, Figure 4 is not necessary. Please consider removing Figure 4.

A: As suggested by the Referee, we understand Figure 4 is not essential, and we decided to remove it.

Q: p. 7/8: Lines 148-156 might be more clearly arranged in a table.

A: The list of physical quantities are now arranged in a table.

Q: p. 10, Figure 8: Please explain in slightly more detail what you mean by '10 different sets of "noisy" input frequencies'.

A: With "noisy" frequencies we mean random frequencies built on top of the original frequencies used in the simulations. The purpose is to generate frequencies which are affected by error, in order to check how the final results of the simulations are affected by the errors in the input data. In order to do that, we draw random numbers from normal (gaussian) distributions, each of them having as mean the original frequency and with a standard deviation equal to 20% of its mean. The definition has been clarified in the manuscript.

Q: p. 10/11, Figure 9a: I don't fully understand the dark bars presented in Figure 9a. While the figure caption suggests that these indicate the average number of fatal collisions, the main text (line 212) suggests that these are the "normalized number of fatal collisions". Please clarify.

A: The dark bars represent the average number of fatal collisions per single realization, which is computed dividing the total number of fatal collisions by the number of realizations. This numbers (dark bars) are between 0 and 1, since a cluster has some finite probability to decompose in each region of the APi, and it cannot decompose more than once. The term "normalized" was used to express that the total number of fatal collisions is divided by the total number of realizations. The text in the manuscript has been clarified.

Q: p. 12, Figure 10: In the figure legend, change "Fragmentation probability" to "De-composition probability".

A: The figure has been corrected.

Q: p. 13, Figure 11a: What is represented by the different curves shown in Figure 11a? The colors 'blue' and 'orange' indicate the presence or absence of the quadrupole electric field but what is represented by the set of curves?

A: The set of curves represent decomposition probabilities for different kinds of clusters. We understand that representing these results for all the kinds of clusters is redundant and not necessary for the scope of the figure (which is used to distinguish between presence and absence of quadrupole electric field), so only the results for one kind of cluster have been plotted now.

СЗ

---

## Author Comment (AC2) · 3 May 2020

Dear Referee #2,

thank you for your comments. Here our answer:

Q: Figure 12 is a central figure in the manuscript: It would be desirable if this figure was discussed more intensively and in more detail in the text. Other illustrations, such as Fig. 8, are less meaningful to be shown graphically and could be deleted without loss of quality.

A: As suggested by the Referee, we understand the explanation of Figure 12 was too concise and we expanded it in more detail in the manuscript. We think that Figure 8 is still worth to appear in the manuscript, since the dependence of the simulation

results on the errors of input data is quite important. Figure 8 shows that the results on decomposition probability slightly change when (even quite large) deviations on the input data are introduced. This means that it is not necessary to know the input data with extreme precision, and it is then possible to "relax" a bit the accuracy in the computation of these data, which often require large computational effort.